# Sustained Treatment Response after Intravenous Cyclophosphamide in a Patient with Therapy-Resistant COVID-19 Acute Respiratory Distress Syndrome: A Case Report

**DOI:** 10.3390/jcm12175506

**Published:** 2023-08-24

**Authors:** Patrick Haselwanter, Christina Bal, Daniela Gompelmann, Marco Idzko, Helmut Prosch, Christian Zauner, Mathias Schneeweiss-Gleixner

**Affiliations:** 1Department of Medicine III, Division of Gastroenterology and Hepatology, Medical University of Vienna, 1090 Vienna, Austria; patrick.haselwanter@meduniwien.ac.at (P.H.); christian.zauner@meduniwien.ac.at (C.Z.); 2Department of Medicine II, Division of Pulmonology, Medical University of Vienna, 1090 Vienna, Austria; christina.bal@meduniwien.ac.at (C.B.); daniela.gompelmann@meduniwien.ac.at (D.G.); marco.idzko@meduniwien.ac.at (M.I.); 3Department of Biomedical Imaging and Image-Guided Therapy, Division of General and Paediatric Radiology, Medical University of Vienna, 1090 Vienna, Austria; helmut.prosch@meduniwien.ac.at

**Keywords:** COVID-19, therapy-resistant ARDS, cyclophosphamide, ICU, interstitial lung disease

## Abstract

Treatment of acute respiratory distress syndrome (ARDS) represents a severe complication of coronavirus disease 2019 (COVID-19) infection and is often challenging in intensive care treatment. Potential positive effects of intravenous cyclophosphamide have been reported in interstitial lung diseases (ILDs). However, there are no data on the use of high-dose cyclophosphamide in therapy-resistant COVID-19 ARDS. We report the case of a 32-year-old male patient admitted to the intensive care unit (ICU) of the Medical University of Vienna due to severe COVID-19 ARDS who required venovenous extracorporeal membrane oxygenation (ECMO) with a total runtime of 85 days. Despite all these therapeutic efforts, he remained in a condition of therapy-resistant ARDS. Unfortunately, the patient was denied for lung transplantation. However, a significant improvement in his respiratory condition was achieved after the administration of an intravenous regimen of cyclophosphamide and prednisolone. After a period of consecutive stabilization, the patient was transferred to the normal ward after 125 days of intensive care treatment. There is a substantial lack of therapeutic options in therapy-resistant ARDS. Our case report suggests that cyclophosphamide may represent a new treatment strategy in therapy-resistant ARDS. Due to its severe adverse effect profile, cyclophosphamide should be used after careful evaluation of a patient’s general condition.

## 1. Introduction

Acute respiratory distress syndrome (ARDS) is a severe complication of coronavirus disease 2019 (COVID-19) infection [1]. Treatment strategies, including lung-protective ventilation and prone positioning, are crucial for therapeutic success. In the case of therapy-resistant hypoxemia, venovenous extracorporeal membrane oxygenation (vv-ECMO) may serve as the last treatment option for either bridging to transplantation or recovery [2,3].

Approaches to immunosuppressive treatment are already common in ARDS treatment [1]. At the beginning of the pandemic, it was thought that infection with severe acute respiratory syndrome coronavirus type 2 (SARS-CoV-2) leads to a massive proinflammatory response characterized by elevated cytokines and chemokines (i.e., cytokine storm), especially interleukin-6 (IL-6) [4,5,6]. Based on these findings, some studies investigated the effect of immunomodulatory therapeutic approaches, like monoclonal antibodies (e.g., tocilizumab for IL-6 signaling) in severe COVID-19 ARDS [4,6,7,8,9]. This approach showed some promising results, at least in a subset of patients [6]. However, SARS-CoV-2 treated with a targeted monoclonal antibody might be too precise, whereas a broader immunosuppressor could be more effective [10]. Large clinical trials show a significant benefit in 28-day survival of COVID-19 ARDS patients treated with glucocorticoids (i.e., 6 mg of dexamethasone) for ten days compared to control groups [11,12].

Cyclophosphamide is an established drug in cancer therapy and even plays an essential role as a rescue drug in rheumatic disorders [13]. Moreover, potential positive effects have been reported in patients with refractory interstitial lung disease (ILD) treated with cyclophosphamide and steroids. However, the field of ILD is heterogenic and can be split into two distinct entities, i.e., idiopathic ILD and ILD with a known etiology. This leads to a broad spectrum of therapeutic approaches in ILD [14,15]. The PANTHER trial did not show any benefits of immunosuppressive treatment in patients with idiopathic pulmonary fibrosis [16]. However, other studies reported a stabilization of pulmonary function in this patient population when treated with intravenous cyclophosphamide compared to azathioprine and oral cyclophosphamide [17,18]. Compared to idiopathic pulmonary fibrosis, an additional immunosuppressive therapy in patients with inflammatory-driven ILDs was reported to have a higher efficacy [19]. There are promising reports about the use of intravenous cyclophosphamide, especially in patients with non-specific interstitial pneumonia [20]. In those patients, the single use of steroids has been reported to be less effective than in patients with a combination of cyclophosphamide and steroids [20]. Similar findings have been reported in patients with systemic rheumatic diseases with pulmonary involvement, such as rheumatic arthritis [21] and perinuclear anti-neutrophil cytoplasmic antibody-positive (p-ANCA) vasculitis [22]. Whereas most of these patients can be treated in an outpatient ward [23], clinicians must be aware of acute interstitial pneumonia, characterized by rapid respiratory deterioration, which frequently requires admission to the intensive care unit (ICU). Diffuse alveolar damage is often detected in these patients, as seen in ARDS [24] or severe COVID-19 infections [25]. Moreover, acute interstitial pneumonia also resembles the clinical course of ARDS and, in radiological imaging, diffuse alveolar damage is represented by ground-glass opacities [1].

An improvement in the respiratory situation in acute interstitial pneumonia with supportive treatment with vv-ECMO was reported in a case series in which one patient was treated with cyclophosphamide and improved under immunosuppressive treatment [26]. The recommendation for intravenous cyclophosphamide is a single-shot application of 600 mg/m^2^ each month in combination with glucocorticoids and should be continued for 3 to 6 months [20]. However, cyclophosphamide is a highly potent drug. It can have severe hepatological [27], pulmonary [28], cardiotoxic [29], and hematological side effects [30] and is only available for off-label use in ILD. Therefore, intravenous cyclophosphamide should be used cautiously only in patients with therapy-refractory ILD.

The use of cyclophosphamide could also have a beneficial impact on therapy-resistant COVID-19 ARDS. A notable use of cyclophosphamide in a 25-year-old patient with glomerulonephritis and COVID-19 infection was reported in 2020. Despite detected ground-glass opacities in a computer tomography imaging scan and a positive SARS-CoV-2 polymerase chain reaction (PCR) test, the patient did not appear with severe respiratory symptoms under ongoing immunosuppressive treatment, including cyclophosphamide [31]. However, there is no report of a patient suffering from severe COVID-19 ARDS treated with intravenous cyclophosphamide. Here, we report the case of a patient with severe therapy-resistant COVID-19 ARDS who showed significant respiratory improvement after treatment with cyclophosphamide.

## 2. Case Presentation

In March 2021, a 32-year-old male patient (body weight: 100 kg, height: 169 cm) appeared eight days before his COVID-19 infection at the trauma-surgery outpatient unit of the Vienna General Hospital with paresthesia in the L3/L4 segments and pain radiating into the left limb. Subsequently, the patient was admitted to the orthopedic ward for further evaluation due to a central disc herniation of the L3/L4 segments. Spinal stenosis was verified on a magnetic resonance imaging (MRI) scan. A SARS-CoV-2 PCR test initially conducted at admission was negative and vital parameters (saturation, blood pressure, temperature) were within the normal range. In his medical history, obesity, depression, adaptation disorder, generalized anxiety disorder, and hepatic steatosis were reported.

Due to a fever of up to 39 °C and a positive COVID-19 PCR test with a cycle threshold of 12.4 (Figure 1), the patient was transferred to the pulmonology ward eight days after hospital admission (day 0). The patient was in a reduced general condition, with a saturation of 91% with no oxygen support, a heart rate of 100 bpm, and blood pressure of 126/79 mmHg. In the initial arterial blood gas analysis, the patient had a pH of 7.46, pCO_2_ of 33.6 mmHg, and pO_2_ of 63.2 mmHg. Rapid respiratory deterioration under non-invasive ventilation (NIV) made admission to the ICU necessary. The patient showed no improvement under high respiratory support, prone positioning, and high-dose dexamethasone. Therefore, mechanical invasive ventilation (MIV) was necessary on day 14. Despite prone positioning and the application of high positive end-expiratory pressure (PEEP), the patient had a pO_2_/FiO_2_ (P/F) ratio under 80 for more than 6 h. Due to severe ARDS, vv-ECMO was implanted on the same day. On day 23, a tracheotomy was conducted based on a suspected long ICU stay with a prolonged weaning period. The patient remained in a condition of therapy-resistant ARDS for weeks with high sweep gas and blood flows. Lung transplantation was denied due to the patient’s obesity.

Initially, antimicrobial therapy was started with ampicillin/sulbactam and switched to cefotaxime and linezolid. After ECMO commencement, antibiotics were initially escalated to meropenem and linezolid. Overall, antimicrobial therapy was adapted several times during his ICU stay due to recurrent ventilator-associated pneumonia. In addition, the patient received antimycotic treatment with fluconazole, which was switched to caspofungin during his ICU stay.

The first trial of 6 mg of dexamethasone for seven days started one day after ICU admission and was tapered for ten days. Thereafter, another trial with 500 mg of prednisolone for 6 days was conducted 17 days after ECMO commencement. The patient received no other immunomodulatory or antiviral therapies like tocilizumab, etanercept, or remdesivir.

The patient received enoxaparin at hospital admission as prophylactic anticoagulation, which was switched to therapeutic anticoagulation with unfractionated heparin during vv-ECMO therapy.

Despite full supportive therapies including optimized ventilation strategies, optimal PEEP trials, prone positioning, and the aforementioned therapies in the ICU, no improvement in the respiratory situation was observed. Even under ECMO therapy (max. sweep gas flow of 6 L/min and max. blood flow of 4 L/min), the P/F ratio remained below 120. However, ECMO support enabled us to apply a lung-protective ventilation strategy with low peak pressure (max. 27 to 29 mbar) and low tidal volumes (136 to 196 mL).

After consultation with the pulmonology department, a bronchoalveolar lavage (BAL) was performed on day 49 (Table 1). The differential cell count showed extended lymphocytosis and neutrophilia (cell count (CC): 1.2 × 10^6^; alveolar macrophages (AM): 0.08; lymphocytes (LYM): 0.42; neutrophil granulocytes (NG): 0.50). At that time, a pulmonary computed tomography (CT) scan showed extensive ground-glass opacities, peribronchovascular consolidations on both sides, and dilatation of the peripheral bronchi (Figure 2). Despite the evidence of dilated bronchi, there was no clear evidence of pulmonary fibrosis. The inflammation profile of the patient’s COVID-19 ARDS resembled non-specific interstitial pneumonia. Therefore, empiric therapy with a single dose of 2000 mg of intravenous cyclophosphamide and 1000 mg of prednisolone was established on day 56 (CP1). Under CP1, the patient’s respiratory condition improved remarkably. A few days after CP1, an increase in tidal volume and P/F ratio (>140) was observed. These improvements in the respiratory situation made it possible to reduce ECMO support (sweep gas flow of 2–4 L/min and blood flow of 2 L/min). At the same time, a slow but significant increase in tidal volume from 196 to 712 mL without an increase in pressure support was observed.

In addition, minute ventilation and tidal volume improved within days (Figure 3). On day 82, a second BAL with a differential cell count showed a reduction in lymphocytes and neutrophils, which could be interpreted as a treatment response (CC: 11.75 × 10^6^; AM: 0.77; LYM: 0.06; NG: 0.15; eosinophilic granulocytes (EG): 0.02). Therefore, the cyclophosphamide and prednisolone regimen was repeated on day 84 (CP2). From CP2 to CP3, the respiratory situation of the patient continually improved. After reducing the sedative medication, the patient showed sufficient spontaneous breathing, which further improved ventilation mechanics. The vv-ECMO support was constantly declining and, consequently, vv-ECMO was successfully explanted with a total runtime of 85 days. After ECMO explantation, the patient had a P/F ratio of 238. Moreover, the entire ventilation effort (PEEP, driving pressure, ventilation mode, peak pressure) was continuously reduced and finally adjusted to a high-flow nasal cannula (HFNC). On day 117, a third BAL showed normal lymphocytes and slightly elevated neutrophils (CC: 2.35 × 10^6^; AM: 0.82; LYM 0.10; NG 0.08), so the cyclophosphamide and prednisolone regimen was continued on day 119 (CP3). On the same day, the patient was decannulated and had adequate respiration. The patient’s general condition improved gradually, so he was transferred to the non-ICU ward on day 125 with 2 L per minute of oxygen supply. Normal lymphocytes and neutrophil granulocytes were detected in the fourth BAL (CC: 10.5 × 10^6^; AM: 0.95; LYM: 0.03; NG 0.01; EG: 0.01). Pulmonary CT scans showed a regression of the peribronchovascular consolidations (Figure 2). Consequently, cyclophosphamide and prednisolone were repeated 62 days after the third administration (CP4).

The patient was continuously in care at Vienna General Hospital due to pulmonary and non-pulmonary complications after his long-term ICU stay (Table 2). The patient received two additional cycles of cyclophosphamide and high-dose steroids, which were generally well tolerated (CP5 and CP6). However, radiological fibrotic pulmonary changes persisted with a chronic dry cough after CP5 and CP6 (Figure 4). Despite early mobilization and physiotherapy, the patient developed an extended critical illness polyneuropathy and additional spinal canal stenosis with peripheral nerve damage and pain syndrome. Moreover, the pulmonary restriction made long-term oxygen therapy with 2 L per minute necessary. A severe critical illness polyneuropathy and peroneal lesion limited the patient’s quality of life more than his pulmonary capabilities. Furthermore, a lung function diagnostic (i.e., body plethysmography and spirometry) was performed two times but discontinued due to the patient’s neuropathic pain.

In a checkup in March 2023, two years after admission, the patient still suffered from symptoms from Long COVID and long-term ICU stay, such as dyspnea, panic attacks, critical illness polyneuropathy, and suspected secondary sclerosing cholangitis. However, the patient was highly committed to lifestyle modification (i.e., physical activity and healthy nutrition) in order to improve his quality of life.

Due to elevated alkaline phosphatase (AP) and γ-glutamyltransferase (GGT) levels, secondary sclerosing cholangitis was suspected, making several hepatological follow-ups necessary. Initially, elevated cholestasis parameters (day of CP3: AP 924 U/L, GGT 1757 U/L and bilirubin 1.4 mg/dL) significantly decreased after ICU discharge. In the latest follow-up, in July 2023, cholestasis parameters were within the normal range (AP 107 U/L, GGT 146 U/L and bilirubin 0.3 mg/dL). In addition, magnetic resonance cholangiopancreatography was conducted, which did not show signs of secondary sclerosing cholangitis.

In a follow-up in August 2023, the patient still suffered from nausea but not from dyspnea, and oxygen support was only necessary during the night. Moreover, symptoms of critical illness polyneuropathy were decreasing. At the time of the last follow-up, a rehabilitation program was still ongoing. The last medications included metoclopramide, ursodeoxycholic acid, pregabalin, trazodone, and lorazepam. However, frequent pulmonological, hepatological, and neurological appointments at Vienna General Hospital are still necessary due to long-term complications after the prolonged ICU stay and severe COVID-19 infection.

## 3. Discussion

COVID-19 ARDS commonly leads to rapid respiratory deterioration, resulting in severe hypoxemia, dyspnea, and bilateral ground-glass opacities in radiological imaging [1]. Overall, COVID-19 ARDS has been connected with a complex immunological reaction leading to a massive cytokine and chemokine response in various organs, especially the lungs [4,5,6]. Upon viral infection, pyroptotic lung cells are thought to induce a massive inflammatory response causing macrophage and T cell activation. In addition, proinflammatory auto-activation of the complement system emerges, which ultimately leads to organ damage and consequently causes respiratory failure and ARDS [32,33]. The central role of hyperinflammation in ARDS might explain the lower effectiveness of antiviral drugs alone in COVID-19 ARDS [33]. In a meta-analysis, the virostatic drug remdesivir reduced mortality in patients with no or low oxygen support. However, there was no benefit in patients with mechanical ventilation [34]. Therefore, therapeutic inhibition of hyperinflammation is crucial in patients with COVID-19 ARDS, leading to lowered mortality [4]. Several therapeutic approaches with monoclonal antibodies and other immunomodulatory therapies brought promising results, at least in a subset of patients [6]. IL-6 inhibitors, such as tocilizumab, significantly decreased mortality [6]. Moreover, promising results were made with glucocorticoids in patients with COVID-19 ARDS [11,12] and in patients with no COVID-19-related ARDS [35]. Especially, dexamethasone and tocilizumab showed a significant reduction in mortality. However, there is a lack of therapeutic alternatives in treating therapy-resistant COVID-19 ARDS.

Besides its use as a chemotherapeutic agent in oncology, cyclophosphamide is a well-established drug in immune-mediated diseases (e.g., systemic lupus erythematosus, p-ANCA-associated vasculitis, severe rheumatoid arthritis, Goodpasture syndrome, minimal change disease) where it is applied to quickly decrease the disease activity and the immune system’s response [36,37,38].

The use of cyclophosphamide in patients with therapy-resistant COVID-19 ARDS could be a new treatment strategy in patients with ARDS and a distinct inflammation profile. A biopsy would be necessary for a distinctive categorization between different interstitial lung diseases, which is often not feasible in the patient’s condition. A beneficial effect of 600 mg/m^2^ of intravenous cyclophosphamide in interstitial lung diseases has already been reported, especially in non-specific interstitial pneumonia, due to their inflammatory reaction. Moreover, the rapid onset of intravenous cyclophosphamide, compared to oral immunosuppressants, such as azathioprine and mycophenolate, is crucial in patients with a therapy-resistant ILD [20,24]. Despite exhausting all therapeutic options, our patient remained in therapy-resistant ARDS without any signs of recovery. The inflammatory profile of the performed BAL resembled non-specific interstitial pneumonia, which could be the reason for the therapy response after cyclophosphamide. A notable reduction in ventilation effort and vv-ECMO support was recognized a few days after the first cyclophosphamide administration. We combined cyclophosphamide with high-dose prednisolone. Moreover, a good therapy response was reported in patients with additional consolidations in high-resolution pulmonary CT scans, as in organizing pneumonia [20,24]. However, the administration of only 6 mg of dexamethasone for ten days had brought no remarkable improvement in the patient’s disease prior to cyclophosphamide treatment. Therefore, a combination of cyclophosphamide and prednisolone might be the reason for the successful treatment.

Early in the beginning of the COVID-19 pandemic, it became clear that SARS-CoV-2-induced lung injury does not present like classic ARDS caused by bacteria [39,40,41]. Based on respiratory mechanics, the perfusion/ventilation mismatch, and the resulting clinical features, COVID-19 ARDS was regarded as an atypical subset of ARDS in the majority of these patients [40,41]. In classic ARDS, pulmonary consolidation and atelectasis lead to an increased shunt volume, resulting in hypoxemia. In addition, impaired lung mechanics cause a decrease in lung compliance and, consequently, lead to respiratory deterioration. Interestingly, COVID-19 patients were found to have a rather preserved compliance [40,41]. Moreover, in COVID-19 ARDS, the shunt volume did not correlate with the fraction of non-ventilated lung tissue. Evidence emerged that the pathophysiological mechanisms behind severe hypoxemia in COVID-19 ARDS were mainly caused by an unusual involvement of the pulmonary microvasculature associated with an impaired perfusion regulation and microvascular coagulopathy [25,40,41,42]. These atypical features of COVID-19 ARDS had affected standard ARDS therapy and led to less effective treatment approaches (i.e., high peep and prone positioning) compared to patients with classic ARDS.

Similar clinical features were detected in our case report, especially at the beginning of his respiratory deterioration. Despite the patient’s high tolerance for NIV and prone positioning, the increasing hypoxemia was unmanageable. After intubation, a drastic decrease in lung compliance was detected, which made lung-protective ventilation without ECMO support unfeasible. Unfortunately, no pulmonary CT scan was performed at the time of intubation. Nevertheless, it is reasonable to state that progressive inflammation within the lung may have caused this deterioration of the lung mechanics.

Some patients who have been infected with SARS-CoV-2 experienced long-term effects from their viral infection. These long-term symptoms/consequences are now known as long COVID and had a major impact on the public health system [43]. Long COVID is poorly understood; different pathophysiological mechanisms are discussed in [44,45]. The symptoms are reported as heterogenous with cardiovascular, pulmonary, neurologic, psychiatric, cutaneous, and further manifestations [43]. Our patient had two risk factors for the development of long COVID (i.e., obesity and a long-term ICU stay) and, indeed, he suffered from various long-term complications including pulmonary, neurologic, psychiatric, and hepatological symptoms [43]. Due to post-COVID-19 ILD (probably caused by the SARS-CoV-2-induced ARDS), the patient suffered from dyspnea and cough and required long-term oxygen therapy. Neurologic complications such as extended polyneuropathy and immobility developed based on a multifactorial etiology (i.e., long-term ICU stay associated with impaired mobilization under ECMO therapy, long COVID, and central disc herniation with spinal stenosis). The last was already present prior to the COVID-19 infection. The patient already had depression before his COVID-19 infection. However, the long-term ICU stay and long COVID could have triggered the patient’s panic attacks. Furthermore, secondary sclerosing cholangitis was suspected based on high cholestasis parameters and was monitored in several hepatological follow-ups. Indeed, SARS-CoV-2 infection itself and long-term ICU stay are thought to be risk factors for the development of secondary sclerosing cholangitis. However, evidence is mostly lacking [46]. Overall, the patient’s long-term complications most probably resemble a multifactorial process.

## 4. Conclusions

Our case of using intravenous cyclophosphamide in a patient with therapy-resistant COVID-19 ARDS as a last resort is worth mentioning and could bring up a possible alternative treatment strategy for therapy-resistant ARDS. Cyclophosphamide is a highly active drug with severe side effects, such as teratogenicity and infertility, and a potential treatment should be considered carefully. However, in patients with therapy-resistant ARDS and lung transplantation as the last option, a prior evaluation of a possible cyclophosphamide administration could represent a feasible treatment option.

However, here, we report just one clinical case where all data were collected retrospectively. Our data represent an observation that might prove a valid hypothesis. For further evaluation of cyclophosphamide in therapy-resistant ARDS, clinical studies are warranted to support our findings and hypothesis.

## Figures and Tables

**Figure 1 jcm-12-05506-f001:**
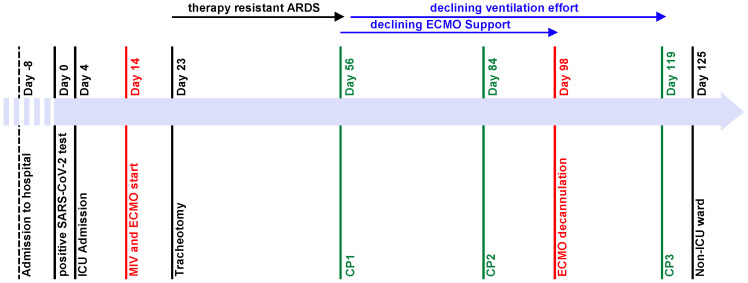
Timeline of major events during ICU stay. The patient was admitted to hospital on day-8. Day 0 is defined as the patient’s first day with a positive COVID-19 PCR test and admission to the pulmonology ward. On day 4, the patient was admitted to the ICU. On day 14, mechanical ventilation was started and a vv-ECMO was implanted on the same day. The patient had no significant improvement of the respiratory situation until CP1 on day 56, gradual improvement of respiratory situation and declining vv-ECMO support after CP1, further declining ventilation effort after CP2 on day 84, and vv-ECMO explantation on day 98. CP3 was administered on day 119, and the patient was transferred to a non-ICU ward on day 125.

**Figure 2 jcm-12-05506-f002:**
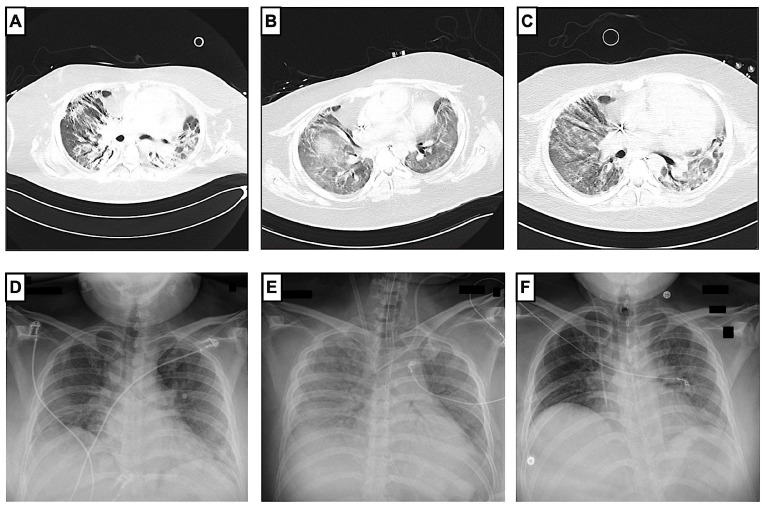
Radiological progression of COVID-19 ARDS before and after administration of cyclophosphamide. Pulmonary CT scans (**A**–**C**) showed a regression of peribronchovascular consolidations and ground-glass opacity during the patient’s hospital stay. First pulmonary CT scan (**A**) was made on day 56, before the administration of cyclophosphamide, and showed findings compatible with organizing pneumonia in the context of COVID-19, with ground-glass opacities, peribronchovascular consolidations, and peripheral expansion of the bronchi. Second CT scan (**B**) was made on day 117 after vv-ECMO explantation and before the third administration of cyclophosphamide. A third CT scan (**C**) was made on day 128 in the non-ICU ward. Chest X-rays (**D**,**E**) show the progression during the ICU stay. The patient had chest X-rays at least once a day as a follow-up. Chest X-ray at ICU admission (**D**) on day 4 showed extended bipulmonary confluent consolidations. A chest X-ray after vv-ECMO implantation on day 14 (**E**) showed extended ground-glass opacity, and another chest X-ray at ICU discharge on day 125 (**F**) showed a regression of ground-glass opacity.

**Figure 3 jcm-12-05506-f003:**
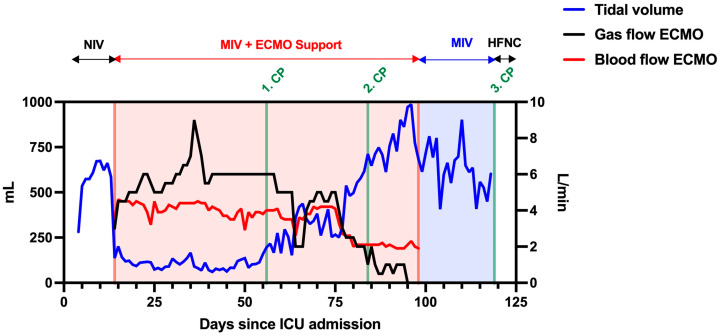
Changes in tidal volume (mL), sweep gas, and blood flow (L/min) of vv-ECMO during ICU stay. The patient had a low tidal volume and high vv-ECMO support until CP1. The vv-ECMO gas and blood flows were constantly decreasing after the CP1, whereas tidal volume was constantly increasing and stabilized after CP2. The total vv-ECMO runtime was 85 days, and the patient had HFNC after CP3.

**Figure 4 jcm-12-05506-f004:**
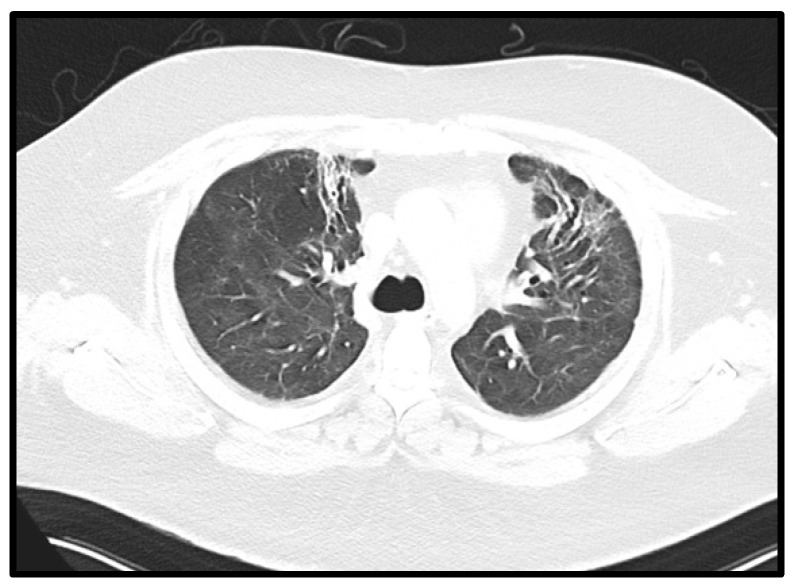
Fibrotic changes in pulmonary CT. Axial CT of the chest performed 31 weeks after onset of symptoms showing reticular abnormalities and traction bronchiectasis in both anterior upper lobes, compatible with post-ARDS lung fibrosis.

**Table 1 jcm-12-05506-t001:** Results of the flow cytometry. A flow cytometry of a BAL fluid was conducted before every administration of cyclophosphamide.

BAL	1	2	3	4	Reference Value
Cell Count (×10^6^)	1.2	11.75	2.35	10.5	9–24 × 10^6^
Alveolar Macrophages (%)	8	77	82	95	>84
Lymphocytes (%)	42	6	10	3	<14
Neutrophile Granulocytes (%)	50	15	8	1	<3
Eosinophile Granulocytes (%)	0	2	0	1	<1
Basophile Granulocytes (%)	0	0	0	0	<1

**Table 2 jcm-12-05506-t002:** Complications and comorbidities during and after ICU stay. The long-term ICU stay and COVID-19 ARDS led to complications detected during and after the patient’s ICU stay. For completeness, pre-existing comorbidities were also included.

Pulmonary Complications	Pre-Existing	During	After	ICU
Ventilator-associated pneumonia		x		
Lung fibrosis		x	x	
Sleep apnea			x	
Non-pulmonary complications	Pre-existing	During	After	ICU
Cyclophosphamide-induced neutropenia		x		
Atrial fibrillation		x		
Cholestasis		x	x	
Secondary sclerosing cholangitis			x	
Critical illness polyneuropathy		x	x	
Peroneal lesion		x	x	
One time grand mal seizure		x		
Panic attacks		x	x	
Comorbidities	Pre-existing	During	After	ICU
Depression	x	x	x	
Migraine	x	x	x	
Obesity	x	x	x	
Disc herniation	x	x	x	

Abbreviations: ICU, intensive care unit.

## Data Availability

No new data were created or analyzed in this study. Data sharing is not applicable to this article.

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
