# Peer review of "Sustained Treatment Response after Intravenous Cyclophosphamide in a Patient with Therapy-Resistant COVID-19 Acute Respiratory Distress Syndrome: A Case Report"

_jcm, 2023, doi:10.3390/jcm12175506_

Round 1
Reviewer 1 Report
Many thanks for the opportunity to read this clinical case that I find interesting. However, some improvements are necessary.
In the case presentation, authors should specify the pandemic period of this clinical case (month,date), the patient's symptoms in the initial infection, the date of first SARS-Co-V2 positivity and the O2 saturation level ad hospital admission. Why was the CT performed so late?Chest CT is usually used in high risk patients. What was the radiologic scores (CT-SS) at the hospital admission? Authors should also specify if other therapies were administred before the cyclophosphamide (antibiotic therapy.. anticoagulants?). Authors should specify, if it is possible, also the therapy at home. I suggest also to add the chest CT images of the fibrotic changes. What are the actual patient conditions? He is still suffering from long-COVID-symptoms? I suggest to divide the discussion and conclusion and to extend both. Authors should start the discussion describing the atypical features of COVID-19 ARDS, after that to describe the current use of cyclophosphamide also for other pathologies. Authors suggest the use of cyclophospamide in COVID-19 ARDS, however authors should also highligh the limitation of this case, because it is only a clinical case (not reproducible). More clinical study are necessary to prove this results.
In the Fig 1, I suggest to change the expression positive COVID-19 test to positive SARS-Co-2 test
Reviewer 2 Report
Thank you for the possibility to review the manuscript titled: “Sustained treatment response after intravenous cyclophosphamide in a patient with therapy-resistant COVID-19 ARDS: A case report”. The manuscript is interesting and easy to read, provides interesting insights about an important topic. The overall quality is high, however, there are several minor recommendations:
-Please change one “is one of the worst” in the introduction section. All of the complications are important and I would not underline one of them.
-Please add some data how did the condition improve over time.
-The discussion section is short and should be expanded. Please add details how ARDS is a complex condition, which involves overactivation of the immune system and the it is regarded as a condition with multiple phenotypes and subphenotypes. Compare the available data of anti-inflammatory therapy in ARDS.
Please take into account the recommendations in the spirit of improving the quality of the submission
Round 2
Reviewer 1 Report
Authors made the requested adjustments. However, I suggest also to enhance the atypical features of ARDS in COVID-19 in the discussion and to describe also long COVID-19 symptoms in order also to extend the discussion
